# Evaluating the Disentanglement of Deep Generative Models with Manifold Topology

**Sharon Zhou, Eric Zelikman, Fred Lu, Andrew Y. Ng, Gunnar Carlsson, Stefano Ermon**
Computer Science & Math Departments, Stanford University
`{sharonz, ezelikman, fredlu, ang, ermon}@cs.stanford.edu,`
` carlsson@stanford.edu`

## Abstract

Learning disentangled representations is regarded as a fundamental task for improving the generalization, robustness, and interpretability of generative models. However, measuring disentanglement has been challenging and inconsistent, often dependent on an ad-hoc external model or specific to a certain dataset. To address this, we present a method for quantifying disentanglement that only uses the generative model, by measuring the topological similarity of conditional submanifolds in the learned representation. This method showcases both unsupervised and supervised variants. To illustrate the effectiveness and applicability of our method, we empirically evaluate several state-of-the-art models across multiple datasets. We find that our method ranks models similarly to existing methods. We make our code publicly available at https://github.com/stanfordmlgroup/disentanglement.

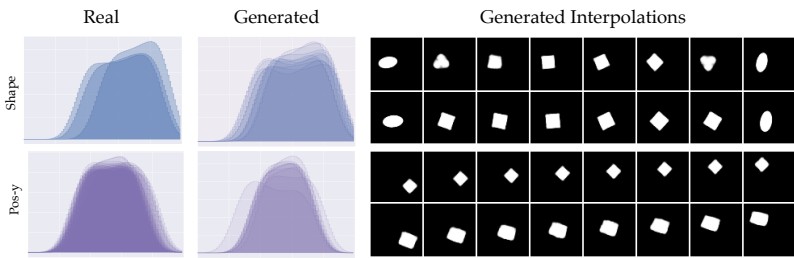

Figure 1: Factors in the *dSprites* dataset displaying topological similarity and semantic correspondence to respective latent dimensions in a disentangled generative model, as shown through Wasserstein RLT distributions—vectorizations of the persistent homology of submanifolds conditioned on a latent dimension—and latent interpolations along respective latent dimensions.

## 1 Introduction

Learning disentangled representations is important for a variety of tasks, including adversarial robustness, generalization to novel tasks, and interpretability (Stutz et al., 2019; Alemi et al., 2017; Ridgeway, 2016; Bengio et al., 2013). Recently, deep generative models have shown marked improvement in disentanglement across an increasing number of datasets and a variety of training objectives (Chen et al., 2016; Lin et al., 2020; Higgins et al., 2017; Kim and Mnih, 2018; Chen et al., 2018b; Burgess et al., 2018; Karras et al., 2019). Nevertheless, quantifying the extent of this disentanglement has remained challenging and inconsistent. As a result, evaluation has often resorted to qualitative inspection for comparisons between models.

Existing evaluation metrics are rigid: while some rely on training additional ad-hoc models that depend on the generative model, such as a classifier, regressor, or an encoder (Eastwood and Williams, 2018; Kim and Mnih, 2018; Higgins et al., 2017; Chen et al., 2018b; Glorot et al., 2011; Grathwohl and Wilson, 2016; Karaletsos et al., 2015; Duan et al., 2020), others are tuned for a particular dataset (Karras et al., 2019). These both pose problems to the evaluation metric's reliability, its relevance to different models and tasks, and consequently, its applicable scope. Specifically, reliance

on training and tuning external models presents a tendency to be sensitive to additional hyperparameters and introduces partiality for models with particular training objectives, e.g. variational methods (Chen et al., 2018b; Kim and Mnih, 2018; Higgins et al., 2017; Burgess et al., 2018) or adversarial methods with an encoder head on the discriminator (Chen et al., 2016; Lin et al., 2020). In fact, this reliance may provide an explanation for the frequent fluctuation in model rankings when new evaluation metrics are introduced (Kim and Mnih, 2018; Lin et al., 2020; Chen et al., 2016). Meanwhile, dataset-specific preprocessing, such as automatically removing background portions from generated portrait images (Karras et al., 2019), generally limits the scope of the evaluation metric's applicability because it depends on the preprocessing procedure and may otherwise be unreliable.

To address this, we introduce an unsupervised disentanglement evaluation metric that can be applied across different model architectures and datasets without training an ad-hoc model for evaluation or introducing a dataset-specific preprocessing step. We achieve this by using topology, the mathematical discipline which differentiates between shapes based on gross features such as holes, loops, etc., alongside density analysis of samples. The combination of these two ideas are the basis for functional persistence, which is one of the areas of application of persistent homology (Cayton, 2005; Narayanan and Mitter, 2010; Goodfellow et al., 2016). In discussing topology, we walk a fine line between perfect mathematical rigor on the one hand and concreteness for a more general audience on the other. We hope we have found the right level for the machine learning community.

Our method investigates the topology of these low-density regions (holes) by estimating homology, a topological invariant that characterizes the distribution of holes on a manifold. We first condition the manifold on each latent dimension and subsequently measure the persistent homology of these conditional submanifolds. By comparing persistent homology, we examine the degree to which conditional submanifolds continuously deform into each other. This provides a notion of topological similarity that is higher across submanifolds conditioned on disentangled dimensions than those conditioned on entangled ones. From this, we construct our evaluation metric using the aggregate topological similarity across data submanifolds conditioned on every latent dimension in the generative model.

In this paper, we make several key contributions:

- We present an unsupervised metric for evaluating disentanglement that only requires the generative model (decoder) and is dataset-agnostic. In order to achieve this, we propose measuring the topology of the learned data manifold with respect to its latent dimensions. Our approach measures the topological dissimilarity measure across latent dimensions, and permits the clustering of submanifolds based on topological similarity.

- We also introduce a supervised variant that compares the generated topology to a real reference.

- For both variants, we develop a topological similarity criterion based on Wasserstein distance, which defines a metric on barcode space in persistent homology (Carlsson, 2019).

- Empirically, we perform an extensive set of experiments to demonstrate the applicability of our method across 10 models and three datasets using both the supervised and unsupervised variants. We find that our results are consistent with several existing methods.

## 2 BACKGROUND

Our method draws inspiration from the Manifold Hypothesis (Cayton, 2005; Narayanan and Mitter, 2010; Goodfellow et al., 2016), which posits that there exists a low-dimensional manifold $\mathcal{M}_{\text{data}}$ on which real data lie and $p_{\text{data}}(\mathbf{x})$ is supported, and that generative models $g : Z \to X$ learn an approximation of that manifold $\mathcal{M}_{\text{model}}$. As a result, the true data manifold $\mathcal{M}_{\text{data}}$ contains high-density regions, separated by large expanses of low-density regions. $\mathcal{M}_{\text{model}}$ approximates the topology of $\mathcal{M}_{\text{data}}$, and superlevel sets of density within $\mathcal{M}_{\text{data}}$, through the learning process.

A $k$-manifold is a space $X$, for example a subset of $\mathbb{R}^n$ for some $n$, which locally looks like an open set in $\mathbb{R}^k$ (formally, for every point $x \in X$, there is a subset can be reparametrized to an open disc in $\mathbb{R}^k$). A *coordinate chart* for the manifold $X$ is an open subset $U$ of $\mathbb{R}^k$ together with a continuous parametrization $g : U \to X$ of a subset of $X$. An *atlas* for $X$ is a collection of coordinate charts that cover $X$. For example, any open hemisphere in a sphere is a coordinate chart, and the collection of all open hemispheres form an atlas. We say two manifolds are *homeomorphic* if there is a continuous map from $X$ to $Y$ that has a continuous inverse. Intuitively, two manifolds are homeomorphic if one can be viewed as a continuous reparametrization of the other. If we have a continuous map $f$ from a

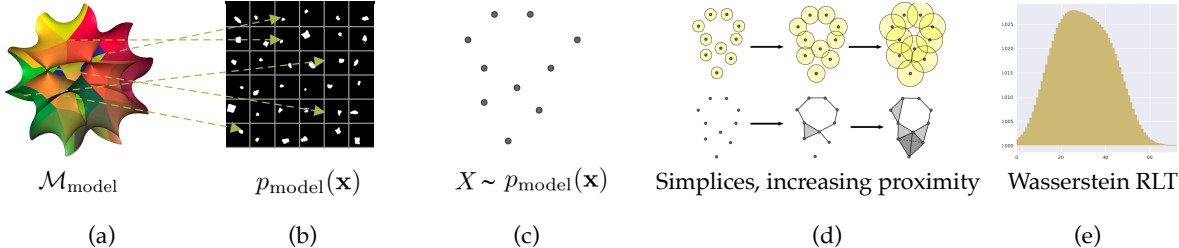

$\mathcal{M}_{\text{model}}$  $p_{\text{model}}(\mathbf{x})$  $X \sim p_{\text{model}}(\mathbf{x})$  Simplices, increasing proximity  Wasserstein RLT

(a)  (b)  (c)  (d)  (e)

Figure 2: Illustration of obtaining Wasserstein Relative Living Times (W. RLTs) from a manifold. (a) a learned manifold with holes, on which (b) $p_{\text{model}}(\mathbf{x})$ is presumed to be supported (images here are generated based on the *dSprites* dataset). From $p_{\text{model}}(\mathbf{x})$, we obtain (c) samples $X$. From $X$, we construct (d) simplicial complexes from increasing the proximity of balls over time, producing a distribution of holes of varying dimensionalities, an RLT. In this example, we first have no holes in the simplicial complex (homology group $H_0$), then both a 1-dimensional hole and no hole ($H_1$,$H_0$), and finally a 1-dimensional hole ($H_1$). From the W. barycenter of many RLTs, we obtain (e) a W. RLT. Fig. (a) is drawn from Hanson (1994) and (c)(d) are drawn from Khrulkov and Oseledets (2018).

manifold $X$ to $\mathbb{R}^n$, and are given two nearby points $\vec{x}$ and $\vec{y}$ in $\mathbb{R}^n$, it is often useful to compare the subsets $f^{-1}(\vec{x})$ and $f^{-1}(\vec{y})$, which are manifolds (where the Jacobian matrix of $f$ is maximal rank). They are frequently homeomorphic, and we will be using topological invariants that can distinguish between two non-homeomorphic manifolds.

Among the easiest topological invariants to numerically estimate is homology (Hatcher, 2005), which characterizes the number of $k$-dimensional holes in a topological space such as a manifold. Intuitively, these holes correspond to low-density regions on the manifold. The field of persistent homology offers several methods for estimating the homology of a topological space from data samples (Carlsson, 2019). Recent work on Relative Living Times (RLTs) (Khrulkov and Oseledets, 2018) has applied persistent homology to generative model data manifolds, also enabling direct comparison of generated data manifolds to real ones. For low-dimensional data (and images), points correspond to their vector representation (flattened matrices of pixels); for high-dimensional data, particularly images, points correspond to vectorized embeddings from a pretrained VGG16 (Simonyan and Zisserman, 2015).

To obtain RLTs, we first construct a family of simplicial complexes–graph-like structures–from data samples, each starting with a set of vertices representing the data points and no edges (Figure 2). These are *witness complexes* that characterize the topology of a set of points, a common method for statistically estimating topological invariants in persistent homology (Carlsson, 2019; Lim et al., 2020). These simplicial complexes approximate the persistent homology of the data manifold by identifying $k$-dimensional holes present in the simplices at varying levels of *proximity*. Proximity is defined with a dissimilarity metric (Euclidean distance) between points. It is used to build a simplicial complex in which a collection of points spans a simplex if all points have proximity measure less than some threshold, as shown in Figure 2d. Varying the threshold gives persistent homology. As proximity increases, simplices are added, creating varying numbers of $k$-dimensional holes, which gives rise to persistence barcodes (Carlsson, 2019; Zomorodian and Carlsson, 2005; Ghrist, 2008).

RLTs are vectorizations of these persistence barcodes, specifically the discrete distributions over the duration of each $k$-dimensional hole as it appears and disappears, or their lifetime relative to other holes. This is merely one method to (partially) vectorize persistence barcodes efficiently, and we leave it to future work to explore alternate methods (Adcock et al., 2013; Bubenik, 2015). To measure the topological similarity between data samples representing two generative model manifolds, Khrulkov and Oseledets (2018) then take the Euclidean mean of several RLTs to produce a discrete probability distribution, called a Mean Relative Living Time; they propose employing the Euclidean distance between two Mean Relative Living Times as the measure of topological similarity between two sets of data samples, known as the *Geometry Score*. Additional background is in Appendix B.

## 3 MANIFOLD INTERPRETATION OF DISENTANGLEMENT

We use prevailing definitions of disentanglement where a disentangled model has a factorized latent space corresponding bijectively to factors of variation (Shu et al., 2019; Higgins et al., 2018; Duan

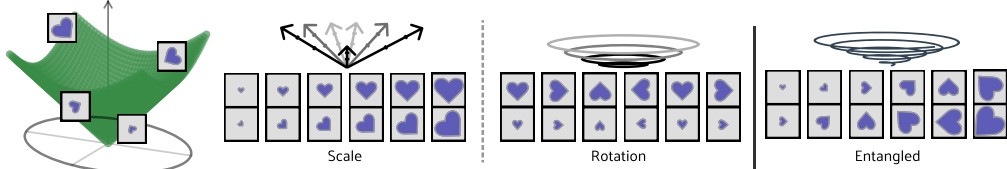

Figure 3: Consider a set of images of a spinning heart with different sizes. We can embed these images on any group that combines a rotational and a scalar invariance (i.e. a group with the SO(2,1) symmetry), visualizing this as a conical shell. The submanifolds conditioned on a given rotation have no holes, while those conditioned on scale have a 1D hole. Notably, the topology of the submanifold when holding scale fixed is different from the topology when holding rotation fixed. Latent dimensions of a disentangled model would embed images on each axis. By contrast, an entangled model may embed one dimension on an axis and the other in a spiral.

et al., 2020). We frame our approach primarily on the definition in Shu et al. (2019), with a more formal connection between their approach and our method in Appendix A.

From the manifold perspective, disentanglement is an extrinsic property that is dependent on the generative model's atlas. Consider a disentangled generative model $g$ with manifold $\mathcal{M}$ that assumes topology $\tau$. We can define another generative model $g'$ with the same underlying manifold $\mathcal{M}$ and $\tau$, but it is entangled and has a different atlas. In fact, we can define several alternate disentangled and entangled atlases, provided there are multiple valid factorizations of the space. As a result, we need a method that can detect whether an atlas is disentangled.

In this paper, we slice $\mathcal{M}$ into submanifolds $U_{s_i=v} \subset \mathcal{M}$ that are conditioned on a factor $s_i$ at value $v$. These conditional submanifolds may have different homology from their supermanifold $\mathcal{M}$. If we observe samples from one factor, e.g. $X_{s_1=v} \sim U_{s_1=v}$ at varying values of $v$, we find that all samples $X_{s_1=v}$ appear identical, except with respect to that single factor of variation $s_i$ set to a different value of $v$. For a generative model, the correspondence between latent dimensions $z_j$ and factors $s_i$ is not known upfront. As a result, we perform this procedure by conditioning on each latent dimension $z_j$.

**Conditional submanifold topology**. For two submanifolds to have the same topology, there needs to be a continuous and invertible mapping between them. First, assume that there exists an invertible mapping, or encoder $e : X \to Z$, and a generative model $g : Z \to X$, where both functions are continuous. Then, for a given $\mathbf{z}$ and $\mathbf{x} = g(\mathbf{z})$, we can recover $\mathbf{z}$ by the composition $\mathbf{z} = e(g(\mathbf{z}))$. We can also construct a simple linear mapping $l : \mathbf{z} \to \mathbf{z'}$, which adapts a factor's value, such that $\mathbf{z'} = l(e(g(\mathbf{z})))$ remains continuously deformable. This holds across factors, where the manifold is topologically symmetric with respect to different factors, i.e. its conditional submanifolds are homeomorphic. As an example, consider a disentangled generative model $g(z_0, z_1, z_2)$ that traces a tri-axial ellipsoid $\frac{x^2}{z_0^2} + \frac{y^2}{z_1^2} + \frac{z^2}{z_2^2} = 1$. If we condition the model on varying values of each factor, the resulting submanifolds are ellipses and have the same topology.

Most complex manifolds have submanifolds that have non-homeomorphic factors of variation. For example, consider a generative model $g(z_0, z_1)$ that traces a cylindrical shell with angle $z_0$, height $z_1$, and for simplicity, no thickness. The submanifolds conditioned on angle $z_0$ form lines (no holes), while the submanifolds conditioned on height $z_1$ form circles (a 1D hole). However, the topology remains the same for a given factor. A visualization of this principle on a cone is shown in Figure 3. Taken together, this means that submanifolds within a factor (intra-factor) are homeomorphic, while submanifolds between factors (extra-factor) can be either homeomorphic or non-homeomorphic.

**Topological asymmetry**. Because topologically asymmetric submanifolds are non-homeomorphic, using a single $e$ that continuously deforms across submanifolds no longer holds under disentanglement. To address this, assume that for each factor $j$, there exists a continuous invertible encoder $e_j : X \to Z_j$ that exclusively encodes information on $j$ from a generated sample. In the cylindrical shell example, this means continuously deforming across submanifolds conditioned on varying values of $z_0$ using $e_0$ (deforming between lines) and likewise for $z_1$ using an $e_1$ (deforming between circles). Note that this formulation prevents continuous deformations between lines and circles. More generally, we cannot continuously deform across submanifolds conditioned by arbitrary factors and expect the topology to be preserved. This procedure now amounts to performing latent traversals along an axis and observing the topology of the resulting submanifolds. In a disentangled model, the $j$-conditional

---

**Algorithm 1:** Procedure for producing W. RLTs on generated images: For generator $g$ with latent prior $\mathcal{P}$ and $N$ samples, $D_z$ latent dimensions, and $n_d$ samples per latent dimension, returns a W. RLT for each dimension. $RLT$ is the RLT procedure from Khrulkov and Oseledets (2018).

---

**for** $i$ in $1 : N$ **do**
    Sample $z^{(i)} \sim \mathcal{P}$ {note: $z \in \mathbb{R}^{N \times D_z}$}
**end for**
**for** latent dimension $d$ in $1 : D_z$ **do**
    **for** $k$ in $1 : n_d$ **do**
      $z' \leftarrow copy(z)$
      Set $z'_d \leftarrow k \sim \mathcal{P}_d$
      Compute $e_z \leftarrow embedding(g(z'_d))$ {e.g. using VGG16}
      Compute $rlt[d,k] \leftarrow RLT(e_z, \gamma = 1/128, L_0 = 64, n = 100)$
    **end for**
    Compute $WB[d] \leftarrow W.Barycenter(rlt[d,k])$
**end for**
**return** $WB$

---

submanifolds exhibit the same topology by continuous composition of $\mathbf{z} = e_j(g(\mathbf{z}))$, using a linear mapping that only adapts factor $j$ across the traversal, i.e. $\mathbf{z'} = l_j(e_j(g(\mathbf{z})))$.

In an entangled model by contrast, more than one factor—such as both the angle and height in the cylindrical shell example—exhibit variation along a dimension $z_j$. Put another way, the topology on submanifolds conditioned on $z_j$ changes when multiple factors contribute to variation along this dimension. Concretely, following the cylindrical shell example, a dimension that encodes height and, after a certain height threshold, also begins to adapt the angle will result in a topology that changes to include a 2D hole. Consequently, submanifolds conditioned on the same latent dimension $z_j$ have the same topology in a disentangled model, yet different topology in an entangled one.

Because we cannot assume that the data manifold of a generative model is completely symmetric, we only consider submanifolds to be homeomorphic along the same factor in a disentangled model. By contrast, since these submanifolds are not homeomorphic in an entangled model, we can measure the similarity across submanifolds to evaluate a model's disentanglement. Using this notion of intra-factor topological similarity, we may sufficiently measure disentanglement in most cases, but it does not shield us from the scenario where a generative model learns a single trivial factor along all dimensions, i.e. a factorization of one. If we assume that there exists assymmetries in the data manifold, then ensuring that the manifold exhibits topological *dissimilarity* between certain factors would disarm that case. We operationalize this by identifying homeomorphic clusters of factors, whereby each cluster has a distinct topology to ensure there is not a factorization of one. Within clusters, we measure topological similarity, but between clusters, we calculate topological dissimilarity. Consequently, topological similarity and dissimilarity form the basis of our evaluation metric. A more principled treatment of how manifold topology measures disentanglement is in Appendix A.

### 3.1 TOPOLOGICAL SIMILARITY USING WASSERSTEIN RELATIVE LIVING TIMES

To estimate the topological similarity between conditional submanifolds, we build on Relative Living Times (Khrulkov and Oseledets, 2018) and introduce Wasserstein (W.) Relative Living Times. Wasserstein distance, unlike Euclidean distance, defines a metric on barcode space (Carlsson, 2019); recall that barcodes are the discrete distributions representing the presence and absence of different $k$-dimensional holes (more formally known as Betti numbers, or $k$-th homology groups), vectorized to form RLTs (Carlsson, 2019). This strongly motivates us to consider Wasserstein over Euclidean distance, which we find empirically to improve separation between distinct factors of variation (on a real disentangling dataset), detailed in Appendix C. Thus, in lieu of the Euclidean mean across RLTs,



Figure 4: Wasserstein RLTs from factors in the *CelebA* dataset, based on the same cluster above, and in distinct clusters below. As one can see, the W. RLTs within a cluster (above) are more similar to each other than to those outside of that cluster (below). Additional examples are in Appendix D.

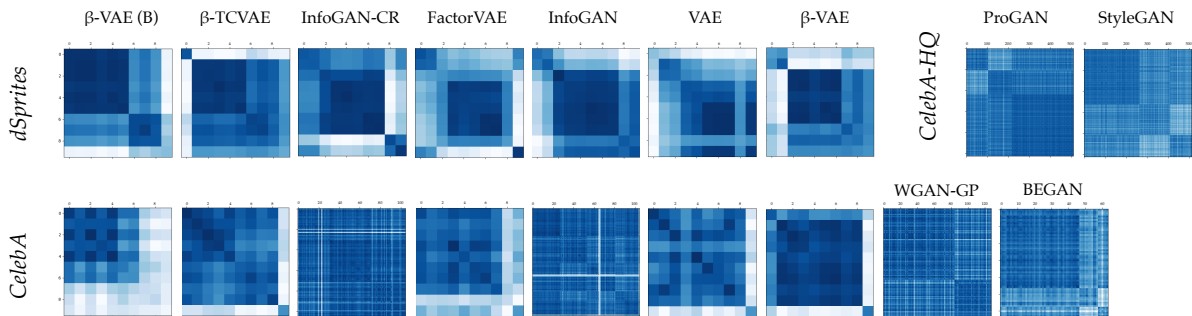

Figure 5: Topological similarity matrices across experimental conditions. Dark clusters along the diagonal indicate homeomorphic clusters. Darker values indicate greater similarity (lower W. distance) to each other. Note that models spectrally cocluster differently across dataset settings.

we equivalently employ the W. barycenter (Agueh and Carlier, 2011). For distances between W. barycenters, we employ standard W. distance.

The W. barycenter $\bar{p}_w$ of the distributions $p_1...p_N$ corresponds to finding the minimum cost for transporting $\bar{p}_w$ to each $p_i$, where cost is defined in W-2 distance: $\bar{p}_w = \arg\min_q \sum_{j=1}^{N} \lambda_j W_2^2(q, p_j)$, where $\lambda \geq 0$ and $\sum_{j=1}^{N} \lambda_j = 1$. This is a weighted Fréchet mean, $\bar{p} = \arg\min_q \sum_{j=1}^{N} \lambda_j d(q, p_j)$, where $d = W_2^2$. In contrast, Euclidean distance, or the $l_2$ norm, is defined using $d = \| \cdot \|_2^2$.

Because our distributions represent discrete unnormalized counts of $k$-dimensional holes, we leverage recent work in unbalanced optimal transport (Chizat et al., 2018; Frogner et al., 2015) that assumes that $p_i$ are not normalized probabilities containing varying cumulative mass. The unbalanced W. barycenter modifies the W. distance to penalize the marginal distributions based on the extended KL divergence (Chizat et al., 2018; Dognin et al., 2019). Unlike Euclidean, Hellinger, or total variation distance, W. distance defines a valid metric on barcode space in persistent homology (Carlsson, 2019).

We provide the procedure for W. RLTs for the generated data manifold in Algorithm 1 and the real data manifold in Algorithm 2 of Appendix I. We also show in Appendix C that the use of both W. RLTs and W. distance result in a distance metric on sets of RLTs that best separates similar and dissimilar topological features, as measured using persistent homology.

## 3.2 EVALUATION METRIC

Equipped with a procedure for measuring topological similarity, we develop a disentanglement evaluation metric from intra-cluster topological similarity and extra-cluster topological dissimilarity. Beginning with intra-factor topological similarity, we are concerned with the degree to which the topology of $p_{\text{model}}(\mathbf{x}|s_i = v)$ varies with respect to a factor $s_i$ at different values of $v$. Specifically, we condition the manifold on a particular factor $s_i$ at value $v$, while allowing other factors $s_{\setminus i}$ to vary. We then measure the topology of this conditional submanifold. For each factor $s_i$, we find the topology of conditional submanifolds at varying values of $v$. A disentangled model would exhibit topological similarity within the set of submanifolds conditioned on the same $s_i$. We visualize similar and dissimilar W. RLTs on factors of the *CelebA* dataset in Figure 4.

For a generative model, the correspondence between latent dimensions $z_j$ and factors $s_i$ is not known upfront. As a result, we perform this procedure by conditioning on each latent dimension $z_j$. We then assess pairwise topological similarity across latent dimensions $\forall_{j,k} d(z_j, z_k)$, where $d$ is the W. distance between W. RLTs. This operation constructs a $j$-dimensional similarity matrix $M$. We use spectral coclustering (Dhillon, 2001) on $M$ to cocluster $z_j$ into $c$ biclusters, which represent different clusters of likely homeomorphic submanifolds conditioned on a shared factor. Spectral coclustering uses SVD to identify, in our case, the $c \leq j$ most likely biclusters, or the subsets of rows that are most similar to columns in $M$. The resulting biclusters create a correspondence from latent dimensions $z_i$ to a cluster of homeomorphic submanifolds conditioned on a factor $h_c$. We then minimize the total variance of intra-cluster variance and extra-cluster variance on the biclusters in $M$ to find the value for $c$. Aggregating biclusters in $M$, we obtain a $c$-dimensional matrix $M'_c$ (see examples in

Figure 5). Using $M'_c$, we compute a score $\mu$ that rewards high intra-cluster similarity $\rho_c = tr(M')$ and low extra-cluster similarity $\rho_{\backslash c} = \sum_a M'_{a \backslash a} - tr(M')$ where $a$ indexes the bicluster rows of $M'$. This score is based on the normalized cut objective in spectral coclustering to measure the strength of associations within a bicluster (Dhillon, 2001). As a result, the unsupervised evaluation metric

$$\mu = \rho_c - \rho_{\backslash c}.$$

**Supervised variant**. In order to capture the correspondence between the learned and real data topology, we present a supervised variant that uses labels of relevant factors on the real dataset to represent the real data topology. This is motivated in large part by F. Locatello et al. (2019), who importantly show that learning a fully disentangled a model requires supervision; further discussion is in Appendix B. While this supervised variant requires labeled data, there are no external ad-hoc classifiers or encoders that might favor one training criterion over another. Persistent homology is computed in the same way for the real data submanifolds as the generated data submanifolds, except that we have desired clusters of factors $s_i$ upfront. See Figure 1 for a comparison between real and generated Wasserstein RLTs of two *dSprites* factors. The major difference is that the generated data manifold is no longer compared to itself, but to the real data manifold, where topological similarity is now computed between the two manifolds: $\forall_{i,j} d(z_j, s_i)$, where $d = W_2$. Note that the relevant factors in the real topology form a specific factorization, so a model that finds an alternate factorization and scores well on the unsupervised evaluation metric may not fare well on the supervised variant.

We use the same spectral coclustering procedure, though this time on a $j \times i$ matrix. Note that because it is not a square matrix, $\rho_c = \sum_{a=0}^{c} M'_{aa}$ and $\rho_{\backslash c} = \sum_{a=0}^{c} \sum_{b=0}^{c} M'_{ab} - \rho_c$, where $c = \min(c, i)$ so we only consider the real factors if $c > i$. Finally, we normalize the final score by number of factors

$$\mu_{\text{SUP}} = \mu/i,$$

to penalize methods that do not find any correspondence to some factors. Ultimately, the supervised evaluation metric favors submanifolds, conditioned on clusters of latent dimensions, which are similar to the conditional submanifolds of the ground truth dataset.

**Limitations**. In Figure 6, we highlight cases where our evaluation metric may face limitations, delineated from scenarios where it would behave as expected. The first limitation is that it is theoretically possible for two factors to be disentangled and, under cases of complete symmetry, still have the same topology. This is more likely in datasets with trivial topologies that are significantly simpler than *dSprites*. While partial symmetry is handled in the evaluation metric with spectral coclustering of homeomorphic factors, complete symmetry is not.

Because we assume the manifold is not exactly symmetric, we do not account for all factors to present symmetry. In order to safeguard against this case, we would need to consider the covariance of topological similarities across pairwise conditional manifolds. This requires selecting fixed points from $v$ that hold two dimensions constant, and subsequently verifying that the topologies do not covary. However, this approach comes with a high computational cost for a benefit only to, for the most part, simple toy datasets. If we assign a Dirichlet process prior over all possible topologies (Ranganathan, 2008) and treat the number of factors as the number of samples, we find the probability of having only a single set of all homeomorphic factors decreases factorially with the number of dimensions $n$.

|  | Similar Topology | Different Topology |
|---|---|---|
| **Entangled** | Expected
Low μ | **Not applicable**
under the manifold
interpretation |
| **Disentangled** | **Unlikely**
in complex data
manifolds | **Expected**
High μ |

Figure 6: Topology-entanglement combinations considered in our method.

An additional limitation of our method is that RLTs do not compute a full topology of the data manifold, but instead efficiently approximate one topological invariant, homology, so that we can comparatively rank generative models on disentanglement. Our overall approach of measuring disentanglement is general enough to incorporate measurements of other topological invariants.

## 4 EXPERIMENTS

Across an extensive set of experiments, our goal is to show the extent to which our metric is able to evaluate across different generative model architectures, training criteria, and datasets. We

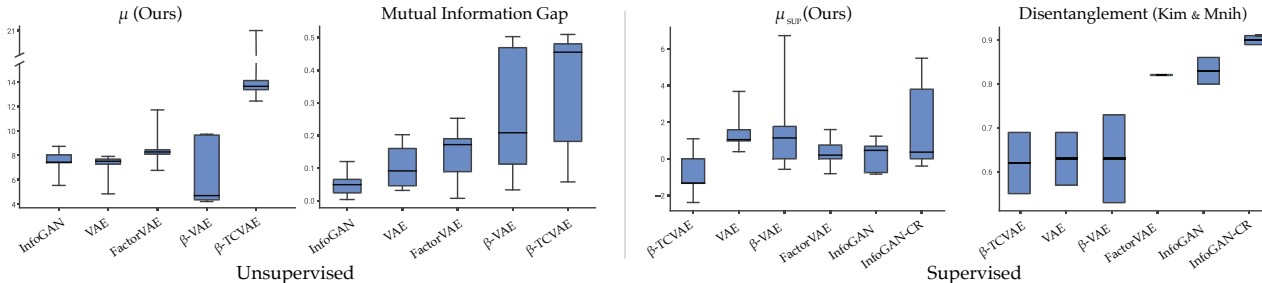

Figure 7: Comparisons of $\mu$ to MIG (Chen et al., 2018b) and $\mu_{\text{SUP}}$ to classifier-based disentanglement score (Kim and Mnih, 2018) on the dSprites dataset.

additionally show that our metric performs similarly to existing disentanglement metrics, without the same architectural or dataset-specific needs. We use pretrained model checkpoints where possible or else train with default hyperparameters. For each model and dataset combination we compute $\mu$ following Algorithms 1 and 3. To compute $\mu_{\text{SUP}}$ we also use Algorithm 2 to generate the W. RLTs of the real dataset. For the embedding function in Algorithms 1 and 2, we used a pretrained VGG16 (Simonyan and Zisserman, 2015) with the last 3 layers removed to embed image samples into 64 feature dimensions. Additional training details are in Appendix G.

**Datasets.** We present empirical results on three datasets: (1) *dSprites* (Matthey et al., 2017) is a canonical disentanglement dataset whose five generating factors {shape, scale, orientation, x-position, y-position} are complete and independent, i.e. they fully describe all combinations in the dataset; (2) *CelebA* is a popular dataset for disentanglement and image generation, and is comprised of over 202K human faces, which we align and crop to be $64 \times 64$ px (Liu et al., 2015). There are also 40 attribute labels for each image; and (3) *Celeba-HQ*, a higher resolution subset of CelebA consisting of 30K images (Karras et al., 2018).

**Generative models.** We compare ten canonical generative models, including a standard VAE, $\beta$-VAE (Higgins et al., 2017), $\beta$-VAE$_B$ (Burgess et al., 2018), FactorVAE (Kim and Mnih, 2018), $\beta$-TCVAE (Chen et al., 2018b), InfoGAN (Chen et al., 2016), InfoGAN-CR (Lin et al., 2020), BEGAN (Berthelot et al., 2017), WGAN-GP (Gulrajani et al., 2017), ProGAN (Karras et al., 2018), and StyleGAN (Karras et al., 2019). We evaluate VAE and InfoGAN variants on dSprites and CelebA, WGAN-GP and BEGAN on CelebA, and ProGAN and StyleGAN on CelebA-HQ. We match models to datasets, on which they have previously demonstrated strong performance and stable training.

**Metric parity.** We find that $\{\mu, \mu_{\text{SUP}}\}$ rank models similarly to several other frequently cited metrics, including: (1) an information-theoric metric MIG that uses an encoder (Chen et al., 2018b), (2) a supervised metric from (Kim and Mnih, 2018) that uses a classifier, and (3) a dataset-specific metric PPL (Karras et al., 2019) that caters to face datasets such as CelebA-HQ. We use scores from their respective papers and prior work (Chen et al., 2018b; Kim and Mnih, 2018; Lin et al., 2020; Karras et al., 2019) to show that our method ranks most or all models the same across each metric ($\mu$ compared to MIG and PPL, $\mu_{\text{SUP}}$ to the supervised method). The source of deviation from MIG is the ranking of $\beta$-VAE; nevertheless, both of our scores exhibit exceptionally high variance across runs, suggesting that $\beta$-VAE has inconsistent disentanglement performance (see Figure 7). The classifier method ranks $\beta$-TCVAE and FactorVAE quite far apart, while ours ranks them similarly. We find that their nearly identical training objectives should rank them more closely and do not find this disparity particularly unexpected. Finally, our method agrees with PPL rankings on CelebA-HQ.

As shown in Table 1, these experiments highlight several key observations:

- Performance is not only architecture-dependent, but also dataset-dependent. This highlights the importance of having a metric that can cater to comparisons across these facets. Nevertheless, we note that $\beta$-VAE$_B$ shows especially strong results on both metrics and two dataset settings.

- As expected, the VAE and InfoGAN variants designed for disentanglement show greater performance on $\mu$ than their GAN counterparts. However, on $\mu_{\text{SUP}}$, we find that BEGAN is able to perform inseparably close to $\beta$-VAE$_B$, suggesting that the model learns dependent factors consistent with the attributes in CelebA.

| Model | Dataset | $\mu$ | $\mu_{\text{SUP}}$ | Model | Dataset | $\mu$ | $\mu_{\text{SUP}}$ |
|---|---|---|---|---|---|---|---|
| $\beta$-VAE$_B$ | dSprites | **23.53** $\pm$ 8.14 | **3.55** $\pm$ 4.25 | $\beta$-VAE$_B$ | CelebA | 4.73 $\pm$ 2.27 | **0.29** $\pm$ 0.25 |
| $\beta$-TCVAE | dSprites | 14.92 $\pm$ 3.46 | -0.79 $\pm$ 1.35 | $\beta$-TCVAE | CelebA | 10.66 $\pm$ 2.48 | 0.04 $\pm$ 0.36 |
| InfoGAN-CR | dSprites | 9.73 $\pm$ 4.03 | 1.85 $\pm$ 2.63 | InfoGAN-CR | CelebA | 0.72 $\pm$ 0.27 | 0.07 $\pm$ 0.15 |
| FactorVAE | dSprites | 8.66 $\pm$ 1.83 | 0.35 $\pm$ 0.90 | FactorVAE | CelebA | 8.53 $\pm$ 4.53 | -0.14 $\pm$ 0.28 |
| InfoGAN | dSprites | 7.42 $\pm$ 1.19 | 0.16 $\pm$ 0.92 | InfoGAN | CelebA | 1.11 $\pm$ 0.81 | 0.00 $\pm$ 0.01 |
| VAE | dSprites | 7.05 $\pm$ 1.25 | 1.54 $\pm$ 1.27 | VAE | CelebA | 6.98 $\pm$ 2.78 | 0.00 $\pm$ 0.15 |
| $\beta$-VAE | dSprites | 6.53 $\pm$ 2.89 | 1.81 $\pm$ 2.90 | $\beta$-VAE | CelebA | **15.10** $\pm$ 8.94 | 0.13 $\pm$ 0.38 |
| StyleGAN | CelebA-HQ | **1.03** $\pm$ 0.24 | **0.77** $\pm$ 0.07 | BEGAN | CelebA | 0.85 $\pm$ 0.25 | 0.22 $\pm$ 0.10 |
| ProGAN | CelebA-HQ | 0.68 $\pm$ 0.08 | 0.37 $\pm$ 0.45 | WGAN-GP | CelebA | 0.83 $\pm$ 0.29 | 0.07 $\pm$ 0.13 |

Table 1: Experimental results on several generative models and dataset settings for our unsupervised $\mu$ and supervised $\mu_{\text{SUP}}$ metrics, across five runs. We find that, consistent with other disentanglement metrics, no model architecture that we evaluated supercedes all others on every metric and dataset. Higher values indicate greater disentanglement.

- With similar training objectives, $\beta$-TCVAE and FactorVAE demonstrate comparable strong performances on $\mu$ across both dSprites and CelebA. $\beta$-TCVAE displays slight, yet consistent, improvements over FactorVAE, which may point to FactorVAE's underestimation of total correlation (Chen et al., 2018b). Nevertheless, FactorVAE demonstrates higher $\mu_{\text{SUP}}$ on dSprites.

- StyleGAN demonstrates consistently higher disentanglement, compared to ProGAN, which supports architectural decisions made for StyleGAN (Karras et al., 2019).

Different datasets and requirements for evaluating disentanglement may favor an unsupervised variant over a supervised variant, and vice versa. Without known factors or without factors of interest, the unsupervised variant is clearly the more favorable or in some cases the only option.

One benefit of using the unsupervised variant even if the factors are known is when multiple combinations of factors are valid to disentangle the data. For example, one can imagine disentangling color into RGB {red, green, blue}, HSL {hue, saturation, lightness}, or HSV {hue, saturation, value}. The unsupervised evaluation metric would be satisfied with a model that disentangles into any of these. A supervised evaluation metric, for which the known factors are RGB, however, would not. RGB is a human (supervised) prior that dictates not only whether but how the factors disentangle. In effect, this leads to differences in rankings across the two variants. Nevertheless, in the case of a disentanglement dataset, such as dSprites, where the factors were created first and the data was created from those factors, we would expect that the known factors are reasonable ones, if not the best ones according to human judgment, to disentangle. That is, while there may be alternatives, this is a good, and quite possibly the most intuitive, human prior.

However, datasets in the wild, which CelebA veers closer to, differ in this respect. We do not know the complete set of factors of variation for CelebA's human faces, and the attributes provided in the metadata are a subset at best. In this case, if one wishes to measure disentanglement generally, we suggest using unsupervised approaches to assess disentanglement of a learned model. If one wishes to measure disentanglement of specific factors, e.g. sunglasses and hair color, then the supervised variant would be more appropriate. Nevertheless, different variants suit different needs, and across datasets and variants, we may expect different models to emerge as superior in their ability to disentangle.

## 5 CONCLUSION

In this paper, we have introduced a disentanglement evaluation metric that measures intrinsic properties of a generative model with respect to its factors of variation. Our evaluation metric circumvents the typical requirements of existing evaluation metrics, such as requiring an ad-hoc model, a particular dataset, or a canonical factorization. This opens up the stage for broader comparisons across models and datasets. Our contributions also consider several cases of disentanglement, where labeled data is not available (unsupervised) or where direct comparisons to user-specified, semantically interpretable factors are desired (supervised). Ultimately, this work advances our ability to leverage the intrinsic properties of generative models to observe additional desirable facets and to apply these properties to important outstanding problems in the field.

ACKNOWLEDGEMENTS

We would like to thank Torbjörn Lundh, Samuel Bengmark, and Rui Shu for their helpful feedback and encouragement in the preparation of our manuscript.

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
