# OpenReview forum: "Evaluating the Disentanglement of Deep Generative Models through Manifold Topology"
_ICLR.cc/2021/Conference — ICLR 2021 Poster_

### Official Review · AnonReviewer4 · 2020-10-27
**Novel application of topological similarity to disentangling metrics, but needs more experimental validation**

**Rating:** 5
**Confidence:** 2

**Review:**

Summary: Introduces unsupervised disentangling metric that measures homeomorphic similarity between submanifolds conditioned on a given factor, and homeomorphic dissimilarity on submanifolds conditioned on different factors. The paper also includes a supervised variant which can directly assess topological similarity of submanifolds with label-spaces. The paper also introduces a novel variation of RLTs that  employs wasserstein distance instead of euclidean distance.

Strengths:
* Novel application of topological similarity to unsupervised evaluation of disentangling.
* Unsupervised and supervised metrics are both in the same units, and can be directly compared.
* Experimental comparison, using the proposed metric, of many standard disentangling models on challenging datasets.

Weaknesses:
* The comparison of the proposed metric to mutual information gap and the metric from Kim and Mnih 2018 shown in Figure 7 is not convincing. Not only does the ranking change, but the distributions of scores look wildly different between models (e.g., \mu has low variance for VAE in both supervised and unsupervised cases, but both MIG and Kim & Minh show relatively high variance for VAE). It is possible that discrepancy is a good thing, and that the proposed model is capturing disentangling even better than the baseline metrics. However, there is not enough experimental validation to know whether this is the case. The paper would be stronger if it had more experimental validation, including a replication of the reported scores from the MIG (Chen et al., 2018) and (Kim and Minh 2018) paper, and deeper analysis explaining the relative differences in scores for models. Without this component, it is difficult to know whether the model comparison results (Table 1) are meaningful.
* It would be nice to include a discussion of how the proposed metric accounts for “dead” units (e.g., Eastwood & Williams 2018)? These might result in topologically similar submanifolds (since the conditional distributions would be nearly identical) and a worse score for the metric, even though the representation is arguably well-disentangled.

Clarity:
* Typo: Page 6, Line 5: “Figure 9” should be “Figure 4”

Other notes:
* It would be very interesting to see an experiment where the unsupervised metric is used to choose hyper-parameters of a model which is then evaluated in a supervised way, to demonstrate that the unsupervised metric is a valid substitute for a supervised metric.

---

> ### Author Response · Authors · 2020-11-16
> **Author response**
>
> Thank you for helpful comments and notes on the novelty and experimental breadth of the work.
>
> We provided a deeper analysis on the scores of the models at the end of our Experiments section, just before the conclusion. Specifically, we've stated that different datasets and requirements for evaluating disentanglement may favor an unsupervised variant over a supervised variant, and vice versa. Without known factors or without factors of interest, the unsupervised variant is clearly the more favorable or in some cases the only option. One benefit of using the unsupervised variant even if the factors are known is when multiple combinations of factors are valid to disentangle the data.
>
> For example, one can imagine disentangling color into RGB \{red, green, blue\}, HSL \{hue, saturation, lightness\}, or HSV \{hue, saturation, value\}. The unsupervised evaluation metric would be satisfied with a model that disentangles into any of these. A supervised evaluation metric, for which the known factors are RGB, however, would not. RGB is a human (supervised) prior that dictates not only whether but how the factors disentangle. In effect, this leads to differences in rankings across the two variants. Nevertheless, in the case of a disentanglement dataset, such as dSprites, where the factors were created first and the  data was created from those factors, we would expect that the known factors are reasonable ones, if not the best ones according to human judgment, to disentangle. That is, while there may be alternatives, this is a good, and quite possibly the most intuitive, human prior.
>
> However, datasets in the wild, which CelebA veers closer to, differs in this respect. We do not know the complete set of factors of variation for CelebA's human faces, and the attributes provided in the metadata are a subset at best. In this case, if one wishes to measure disentanglement generally, we suggest using unsupervised approaches to assess the disentanglement of a learned model. If one wishes to measure the disentanglement of specific factors, e.g. sunglasses and hair color, then the supervised variant would be more appropriate. Nevertheless, different variants suit different needs, and across datasets and variants, we may expect different models to emerge as superior in their ability to disentangle.
>
>
> Regarding the “dead” units, as discussed in Eastwood & Williams 2018, these are clustered under our method based on topological similarity, and therefore not penalized in our evaluation metric as we use the trace on the clusters in the computation (and not, for example, their mean). Arguably, however, the resulting representation would be more compact ("better" in some sense, though not necessarily disentangled), if there are no or fewer dead units.
>
>
> Agreed on the future work direction and a good extension, as the StyleGAN line of work has similarly done so with their PPL evaluation metric, incorporating it into the training of StyleGAN2, a subsequent iteration of their work.
>
> Thanks again for your helpful comments. We hope these points help elucidate your questions.

---

### Official Review · AnonReviewer3 · 2020-10-28
**Disentanglement metric measuring intrinsic properties of a generative model with respect to its factors of variation.**

**Rating:** 8
**Confidence:** 4

**Review:**

The paper presents a disentanglement metric to measure the intrinsic properties of a generative model with respect to the factor of variation in the dataset. Toward this, the paper first assumes disentangled factors reside in different manifolds. These different manifolds are the sub-manifolds of some manifold M for a given disentangled generative model. The paper considers the fact that in an entangled model the sub-manifolds are not homeomorphic and thus similarity across submanifolds can be measured to evaluate a model’s disentanglement. As such, disentanglement is related to the topological similarity.  For measuring topological similarity, the paper then introduces Wasserstein Relative Living Times. The proposed metric is used to evaluate standard disentanglement methods and datasets demonstrating the importance.

Pros:
- The paper is well written and the proposed metric is well articulated.
- The manifold interpretation of disentanglement (section 3) is clear and could be considered a stand-alone contribution to the disentanglement community.
- The experiments covered depth in terms of dataset selection and the number of models considered for the evaluation of the metric.

Few questions/suggestions:
- Can the authors briefly describe why W. Distance defines a valid metric on barcode space and others don’t, or point out to relevant literature in section 3.1?
- In Li et. al., 2019, MIG-sup was proposed to remedy the weakness of the MIG metric. For the unsupervised portion, since MIG was considered a comparision, can authors also compare their method with MIG-sup? Or at least discuss it along with MIG?

Li, Z., Murkute, J. V., Gyawali, P. K., & Wang, L. (2019, September). PROGRESSIVE LEARNING AND DISENTANGLEMENT OF HIERARCHICAL REPRESENTATIONS. In International Conference on Learning Representations.

(Update): The score has been updated after the rebuttal phase.

---

> ### Author Response · Authors · 2020-11-16
> **Author response**
>
> Thank you for your enthusiastic response on the clarity of the manifold interpretation of disentanglement, depth of experiments, and clear writing.
>
> Regarding W. distance defining a valid metric on barcode space, thanks for pointing that out. We added the appropriate citation to Carlsson (2019) and moved the citation closer to the phrase in another area, both occurring in Section 3.1. Intuitively, W. distance can be seen as defining an edit distance in terms of a penalty function in barcode space, and thus satisfies the triangle inequality. A precise definition can be found in Section 5.1 of “Persistent Homology and Applied Homotopy Theory” (Carlsson, 2019).
>
> Regarding the Li et al., 2019 paper, we have cited it in the main text of our background and related work sections (and an additional mention in the appendix), as it is very pertinent. Following ICLR guidelines that allow small contained studies during the revision period, we have run such a study for MIG-sup VAE, β-VAE, and β-TCVAE. We found that for these models, MIG and MIG-sup are highly correlated (r2=0.95) and plot a box-plot over several run in Appendix G. Based on that paper (Li et al., 2019), MIG-sup and MIG correspond, respectively, to the disentanglement concepts of consistency (when one dimension exclusively encodes multiple factors) and restrictiveness (when one factor is exclusively split into multiple dimensions) in Shu et al. 2019, to which we had connected in Appendix A. We included this note about MIG-sup in that section (Update: We performed several runs for a more complete analysis of MIG-sup).
>
> Thank you again for your kind response and hope these points help elucidate your questions.

---

> > ### Comment · AnonReviewer3 · 2020-11-21
> > **Thanks for the update.**
> >
> > Thanks for your response and the extended experiments on disentanglement metrics.

---

### Official Review · AnonReviewer2 · 2020-10-28
**Official Blind Review #2**

**Rating:** 7
**Confidence:** 3

**Review:**

This paper proposes a new metric to quantify disentanglement by leveraging ideas from manifold topology, more precisely the homology of conditional submanifolds. It has a strong theoretical grounding (clearly better than existing metrics) and results appear to be promising.

However it is at times quite hard to follow, and I am not entirely confident that I fully understood how the Wasserstein RLTs were estimated in practice. I’d still recommend publication, given the novelty and importance of a theoretical sound metric for disentanglement.

Comments/questions:
1. The Introduction, Background and Manifold interpretation sections do a good job of introducing the subject, covering the literature and providing enough background about differential geometry, but I think a clear algorithmic demonstration of steps (d) and (e) of Figure 2 is lacking.
      1. How exactly do you estimate the Wassertein RLT and compute the matrix M?
      2. Writing the exact equations used, e.g. in the Appendix, would be helpful. Overall I found the main text to rely on explanations a bit more than would be necessary, where some math would be clearer (having the code will be most helpful however)
      3. The actual disentanglement metrics $\mu$ and $\mu_{sup}$ might deserve a \begin{equation} for clarity.
      4. You never explicitly say if “high $\mu$ => high disentanglement”, which I think is how one should interpret Table 1 given the main text?

2. Providing a bit more explanations about how to interpret the RLT histograms would be valuable. I imagine that if the histograms vary as a function of their conditioning, this indicates entanglement?

3. Figure 7 was interesting, but the specific mention of a mismatch in how the classifier method (i.e. Disentanglement (Kim & Mnih) I assume?) ranks $\beta$-TCVA and FactorVAE differently was hard to follow (i.e. these do not appear that close in your $\mu_{sup}$ figure?)

4. Do the topological similarity matrices in Figure 5 provide interesting information about which factors are grouped together?
      1. Could you add labels on the axes and discuss what each cluster represents in a few selected models/datasets?

5. How does the method relate to what was done in GEOMANCER [1]? They also leverage differential geometry and holonomy (but as a learning signal for representation, not as a metric obviously), so discussing it might be valuable?


[1] https://arxiv.org/abs/2006.12982

---

> ### Author Response · Authors · 2020-11-16
> **Author response**
>
> Thank you for your helpful comments and suggestions that have helped with clarifying the procedural steps and the figures. This is important to making our work easier to digest in visuals and algorithms separate from the main text, and we appreciate these suggestions.
>
> For clarifying the precise steps of computing W. RLTs and the matrix M, we add the central algorithms to Appendix I and refer to them in the main text. Specifically, we break down the full procedure into three simpler algorithms: W. RLT procedure on the generated data manifold, W. RLT procedure on the real data manifold, and evaluation metric μ calculation. We have linked to the codebase in our abstract.
>
> We have also included additional explanations in the figure captions to help with their clarity. Specifically, we have revised these 3 requested items: (a) RLT diagrams can be interpreted as vectorizations of the persistent homology of, in this case, a conditional submanifold, providing information about the homology groups (k-dimensional holes and their relative lifetime as proximity, or the radius of the ball around each point, varies) as estimated by persistent homology. We add this clarification in the text and captions. (b) we clarify that Figure 7 is on the dSprites dataset and provide further discussion of observed differences at the end of the experiments section. The mismatch that we believe you refer to focuses on the results in Table 1, less so on Figure 7, so we switch their ordering to help with reading that section). (c) We note that the largest clusters in dSprites are consistently orientation, which is consistent with the real disentangling factors, as they have the most values, and shape which has the least values generally corresponds to the smallest cluster. However, since Figure 5 shows the unsupervised variant and these models do not perfectly disentangle, one cannot assign a corresponding factor without a valid correspondence being evaluated.
>
> Regarding the related work on Geomancer, we note that holonomy and homology are separate concepts, yet certainly related. While homology is a topological invariant of the manifold, holonomy used in the Geomancer paper hinges on differential geometry methods and could be an interesting subject for future work, given the well known connections, such as the Gauss-Bonnet theorem,  between differential geometric quantities such as curvature and homological invariants. We include this in our extended background section, as it is a relevant piece of work at this intersection to reference and to bolster.
>
> Finally, stylistically, we add \begin{equation} to the evaluation metric equations. We also explicitly highlight that a higher μ entails greater disentanglement in the Table 1 caption and in the main text.
>
> Thank you again for your comments that helped clarify aspects of the paper.

---

### Official Review · AnonReviewer1 · 2020-10-30
**Interesting looking paper**

**Rating:** 6
**Confidence:** 1

**Review:**

Summary

The paper proposes a novel metric for evaluating disentanglement by taking a manifold-topological perspective on the representations learnt. The key insight is that for a disentangled representation, when we fix a certain factor of variation at different values the topology of the conditional sub-manifolds should be similar. Using this insight the paper proposes a metric for disentangling which does not require annotations of the factors of variation and is more general than previous such tests.

Strengths
+ Having an approach that is general and easy to compute across datasets and models makes a lot of sense

Weaknesses

It would be nice to further clarify the intuitions for how disentangling relates to the manifold structures using more examples in the paper for people who are not familiar with the manifold topology literature. For example, in Fig. 3 it would be nice to show qualitatively what an entangled model would do for the rotation scale disentangling situation.

It seems somewhat surprising that \beta-VAE_{B} does so much worse than \beta-VAE on CelebA, were the hyperparameters tuned separately for the CelebA dataset?

---

> ### Author Response · Authors · 2020-11-16
> **Author response**
>
> Thank you for your comments and note on a general simple evaluation metric — this is exactly our aim to contribute this to the literature.
>
> Regarding your question on Fig. 3 of the entangled situation: an entangled model would qualitatively embed the hearts in a spiral around the conical shell; this spiral would correspond to hearts that both change in size and in rotation. For this to be clear to future readers, we have included a visual example in that figure.
>
> The hyperparameters were the same across datasets, not tuned separately, and specifically default from their respective papers (Appendix G). Such discrepancies can be seen across datasets (one architecture is often not globally best), which was our motivation to ensure that experiments were not conducted on a single dataset, but multiple, spanning toy disentanglement datasets like dSprites and more "in-the-wild" datasets like CelebA.

---

### Official Review · AnonReviewer5 · 2020-11-06
**An exciting demonstration of topology-based assessments of generative models**

**Rating:** 5
**Confidence:** 5

**Review:**

# Synopsis of the paper

This paper presents a new model for assessing to what extent a deep
generative model is disentangled. In contrast to existing methods, this
paper proposes an approach founded on topological considerations: by
assessing the topological dissimilarity of submanifolds of a given model,
which are conditioned on an individual factor. The overarching idea is
that a fully-disentangled model should exhibit high topological
dissimilarity for *different* factors, while exhibiting low topological
dissimilarity for the *same* factors (but with different values for said
factors).

These topological dissimilarity assessments are achieved by employing
persistent homology, a method from topological data analysis that
generalises ordinary simplicial homology (i.e. 'counting
high-dimensional holes') to a multi-scale setting (in which point clouds
are being assessed). Previous work for GAN comparison, viz. the
'geometry score' is extended by a new internal metric, the Wasserstein
distance.

A set of experiments demonstrates the utility of the proposed approach,
validating previous knowledge.

# Summary of the review

This paper is tackling a highly relevant topic (disentanglement
analysis) and provides a novel, fresh perspective that is mathematically
well justified. The paper is chock full of excellent ideas that make me
excited about their potential; I am sure that this will be shine a light
on the potential of topology-driven methods.

However, while the paper is well written for the most part, I cannot
fully endorse it for publication yet because of two reasons:

1. The paper is missing crucial details about its comparison method.
   While I know the existing work to piece everything together, I am
   not sure the method as such is reproducible at the moment.

   I am agreeing with most of the claims in the paper, but I would be
   hard-pressed to implement exactly the same method in the same
   fashion, because not all the details are there.

2. While most of the paper flows very well, in an almost conversational
   style that I very much appreciate, there are also several places at
   which concepts are not explained correctly or with sufficient
   precision. There are also some issues with the terminology that make
   it harder to fully understand what is going on.

If these two points were to be rectified in a revision of the paper,
it would definitely strengthen the message of this paper!

Please see below for detailed comments.

# Detailed comments

- The introduction is really 'on point' and makes an excellent case for
  requiring more foundational approaches

- The way 'topology' is introduced in the introduction is very
  imprecise; I find the discussion of 'high-density' and 'low-density'
  to be problematic here. Algebraic topology, which is where persistent
  homology originated from, has no notion of density; all considerations
  of 'holes' are based on purely algebraic arguments. While it is true
  that you can incorporate density information into persistent homology,
  this discussion is slightly misleading.

  I would point out rather that holes are typically considered to be the
  boundaries of these low-density regions (if the 'density argument' is
  left in there).

- Some of the terms are overloaded and used multiple times: 'latent
  dimension' is used equivalently to 'latent factor', it seems. I would
  strongly suggest to use a unified terminology.

- The term 'homeomorphic dimensions' is vague. I also have to point out
  that the method used in this paper is only *invariant* under
  homeomorphism! More precisely, two topological objects can have the
  same homology (or persistent homology, or barcode, etc.) even though
  they are **not** homeomorphic. This is a crucial detail that is not
  handled correctly in the paper at present; for example, Figure
  5 claims that 'homeomorphic groups' are shown, but the approach cannot
  detect this. If anything, the case should be made for depicting that
  these groups are 'likely' homeomorphic. Alternatively, one could write
  that the proposed approach assesses the *topological similarity* of
  models.

- Often, the term 'topology' is also used as an equivocation.
  Unfortunately, the term is overloaded in mathematics as well: a space
  can *have* a topology (i.e. systems of open sets etc.), but I have the
  impression that the paper often means to say 'shape' or 'topological
  features' (connected components, cycles, etc.) rather than 'topology'.

- The explanation of a manifold could be improved; why not mention the
  definition in terms of 'locally looks like some $\mathbb{R}^d$'? The
  discussion of 'open discs' might leave some readers baffled.

- I disagree with the sentence 'Under the manifold hypothesis, the latent
  space of a deep generative model has an extremely dense underlying
  manifold with few, if any, holes, [...]'. Holes characterise the shape
  of a manifold; the manifold hypothesis only posits the hypothesis that
  data originate from a manifold. It does not say anything about the
  *shape* of the manifold itself, and I do not agree with the notion
  that having no holes is somehow 'better'.

  A generative model that 'learns' that all samples can be arranged
  along a cycle in some latent space can be very appropriate depending
  on the data set.

  (Moreover, Figure 2 in some sense tells the opposite story of this
  sentence.)

- The definition/introduction of (persistent homology) at the end of
  Section 2 could be improved: the Vietoris--Rips complex goes back to
  a 1927 paper by L. Vietoris (https://link.springer.com/article/10.1007/BF01447877);
  an excellent introduction to modern topological concepts is provided
  in the book 'Computational Topology: An Introduction' by Edelsbrunner
  and Harer.

  I also disagree with homology 'approximating the topology'. Homology
  is *one* specific invariant in algebraic topology. In this sense, it
  does not approximate anything but describes a data set in terms of
  its topological features.

- The term 'symplectic vertex' is not correct here, as far as
  I understand. Everything is discussed in terms of simplicial
  complexes; I do not see the connection to symplectic topology here.

- The explanation of the Vietoris--Rips complex could be improved.
  I feel that the explanation in terms of creating edges based on the
  proximity criterion will leave most readers baffled. I would suggest
  adding a brief explanation/illustration about this, maybe using Figure
  2d.

- What is meant by having a manifold 'assume topology $\tau$'? Does this
  refer to the manifold expressing a certain a set of barcodes?

- In Figure 3, I would rather write '[...] the submanifolds conditions on
  a specific rotation [...]'. Upon first reading the figure,
  I misunderstood its central point, namely that *if* the rotation is
  fixed, so that only the scale varies, the resulting submanifold will
  not have any holes, as the variation in scale provides a simple 1D
  structure.

- I would not say that submanifolds have different 'topologies' than
  their super-manifold; rather they might/will have different homology
  maybe or 'shape'?

- While I fully agree with the thrust of the method, I would need more
  details on p. 4. How is the partitioning of the manifold achieved in
  practice? How are the factors chosen? How are their respective values
  chosen? Is it possible to provide more examples here?

- When discussing conditional submanifold topology, consider mentioning
  how these manifolds are being obtained in practice, maybe by providing
  more examples.

- I do not understand the main thrust of the 'Topological asymmetry'
  section. Is the main argument that manifolds conditioned on the same
  factor should have the same shape and, moreover, that manifolds coming
  from the same factor should be more similar to one another than
  manifolds coming from different factors?

- What does it mean that factors are 'not homeomorphic' in Figure 4? Is
  the ground-truth topology known?

- The statement '...that best separates similar and dissimilar
  topologies...' should again be rephrased and made more precise, for
  example by adding '...topological features, as measured using
  persistent homology...'.

- In the 'Metric' section, I would caution strongly against this term.
  The RLT/Wasserstein calculations might result in a metric in the
  mathematical sense (for sure, the Wasserstein distance constitutes
  a metric on the space of barcodes, as the paper mentions), but I am
  not sure whether the *score* calculated in this section constitutes
  a metric. If anything, this would have to be proven.

- The 'Metric' section is also hard to read because of differences in
  terminology. Why is $\xi$ used to represent an index, whereas other
  Greek letters are used for functions? What is $\delta$? Is it the
  proposed measure of disentanglement?

- After the 'Metric' section, I am missing a concluding section, that
  summarises how the method works in practice. The following questions
  need to be answered:

  1. How are topological features computed (i.e., which point cloud is
     being used and which filtration)?

  2. How is the Vietoris--Rips complex computed in practice? How is its
     scale parameter $\epsilon$ chosen?

  3. What is the dimensionality used to compute topological features,
     i.e. are high-dimensional features being used at all?


- '...to those of the reals.': what does this mean?

- It seems that $\tau$ is not used consistently; is $\rho$ meant in
  Figure 6?

All in all, this paper makes me very excited about the potential of
topology-based methods, and I envision that, with some changes, this
will be a strong publication in the future!

# Style & clarity

The paper is well-written; here are some additional suggestions for
further improving its clarity:

- 'generated dSprites' --> 'generated from the dSprites data set'?

- 'by composition' --> 'by the composition'

- The term 'group' is unfortunately somewhat overloaded as well; is
  'factor' more appropriate in some places? Or is the paper actually
  talking about manifold groups?

- I would strongly suggest to check whether 'persistent homology' might
  not be ore appropriate in many places instead of 'homology'. The
  latter should be reserved to the setting of *one* simplicial complex,
  but the main argument of the paper is that one has to adopt
  a multi-scale description of the data set.

# Update after initial revision

I have re-read the revised version multiple times now, and I thank the authors for the amount of work they put into their revision. I am raising my score for the next discussion round. That being said, I want to point out why I cannot fully endorse this paper yet. First, from the point of topological data analysis, the central algorithm is a comparatively small extension of the Geometry Score paper. I agree that this is a superb idea, yet the main contribution for me lies in the application of that technique to disentanglement—and for this to be fully understandable, some more work is needed. For example, putting the central algorithmic details into the appendix will make the adoption of the method that much harder.

Moreover, while I appreciate the overall story and description of the method, I do not think that readers will understand how this disentanglement is actually _achieved_ by means of the proposed TDA approach. I would therefore prefer to see a more 'technical' or 'algorithmic' description of the contributions in the main paper, in particular since I think that the ideas of conditional submanifold topology require more attention.

My expertise is more the topology and less so the application of disentanglement; nevertheless, this paper strikes me as highly ambitious with a lot of potential, yet somewhat unfinished in its present form. I do believe that it has the potential to be extremely impactful with some additional modifications (concerning conciseness, but also experimental details).

I fully realise that this is not yet the desired outcome for the authors; I shall endeavour to discuss this further with my fellow reviewers to see that we can reach a consensus!

---

> ### Author Response · Authors · 2020-11-13
> **Author response: revisions for clarifying procedures and tightening language**
>
> Thank you for your thoughtful and detailed response. We really appreciate the constructive comments therein, as well as the positive praise on the "novel, fresh perspective that is mathematically well justified". Incorporating your suggestions has bolstered our work, and we hope the revisions can improve the assessment of our work.
>
> We've prepared a response, and modified the manuscript, to each of your points. In particular, (1) we provide greater detail and more concrete examples to improve the paper's reproducibility, including certain details from prior work, which we make sure to include in either the main text or Appendix, and (2) we tighten the language on topology and resolved vocabulary ambiguities, particularly for a reader of topology expecting more mathematical rigor.
>
> When addressing mathematical rigor and discussing topology in this context, we acknowledge that we walk a fine line between perfect mathematical rigor on the one hand and concreteness for a more general audience on the other. We hope that we have found the right level for the machine learning community.
>
> We have included these revisions in the updated manuscript.
>
> Regarding (1), we provide the following details on the approach in section 3, the Metric section 3.2, and Appendix G. We also confirm that your interpretation is correct of the Topological asymmetry section in section 3.
>
> *RLT procedure*
> The method for computing the relative living times (RLTs) originates from the Geometry Score paper (Khrulkov et al. 2018) and we include the requested details on their method in the Appendix. Specifically, they use the Gudhi library and compute persistence intervals in a dimension by constructing witness complexes. The witness complex is computed for all filtration values at once to compute a persistence diagram, which summarizes the homology for all ϵ. Topological features are usually not computationally tractable in high dimensions, so we do not use high-dimensional features there — specifically, we are only using k=1 dimension, per the Geometry Score implementation.
>
> *Conditional submanifold in practice*
> The factors of a generating dataset such as dSprites, and their values, are provided with the dataset. Features in the dSprites dataset include shape, orientation, x-position, and y-position. An example image (a data point) is a heart rotated 90 degrees in the top right corner. The values of each feature (factor) are provided in this generating dataset and, in this case, discretized. In generating a submanifold, we would hold a factor, such as orientation, constant, while varying the others (sampling different values for the others) to create a subsample that we then use in the RLT procedure. In a non-toy dataset, such as CelebA, where the factors and their values are not known to high accuracy, we can only estimate a possible subset. In this case, we follow prior precedent and use the binary attributes provided in the dataset, such as wearing sunglasses or black hair color. This type of selection of factors and values are common in disentanglement literature; we do not introduce a novel protocol with the factor selection here.
>
> For a generated manifold, we do not know the factors corresponding to the latent dimensions upfront. As a result, we hold latent dimensions constant and randomly sample values from the latent prior (spherical normal) within a dimension to hold constant while varying the values of others through random sampling. Each set of latent dimension values correspond to a point in the data manifold, which we use the corresponding generative model to generate. These points on the generative model's data manifold are then embedded using an ImageNet-pretrained VGG16 network as a feature extractor (these details are currently in Appendix G). These embeddings produce point clouds from which the persistence barcodes are computed and vectorized using the RLT procedure.
>
> We clarify these in the text and an added algorithm diagram in the Appendix.
>
> Regarding (2), we unify terminology for the following suggestions and updated the manuscript to reflect this by:
> * Clarifying the use of topology throughout the paper and defining it more precisely. Specifically, we include in the Introduction: We achieve this by using topology, the mathematical discipline which differentiates between shapes based on gross features such as holes, loops, etc., alongside density analysis of samples. The combination of these two ideas are the basis for functional persistence, which is one of the areas of application of persistent homology.
> * Using "latent dimensions" throughout for consistency, removing the phrase “latent factor.”
> * Moreover, “homeomorphic dimensions” is replaced with references to “homeomorphic submanifolds conditioned on a factor” and “likely homeomorphic” is used where appropriate.
>
> 1/2 (Continued in next comment)

---

> > ### Author Response · Authors · 2020-11-13
> > **Author response continued: revisions for clarifying procedures and tightening language**
> >
> > 2/2 (Continued from previous comment)
> >
> > * Improving the definition of a manifold to not only discuss "open discs" more formally, but also that it "locally looks like some R^k"
> > * Clarifying discussion of the manifold hypothesis, because the current wording seems confusing, suggesting having no holes (a certain shape) is better, which is not the intention of the sentence.
> > * Rewording our explanation of proximity and persistent homology as: Proximity is defined using a dissimilarity measure between points. It is used to build a simplicial complex in which a collection of points spans a simplex if all points have proximity measure less than some threshold. Varying the threshold gives persistent homology.
> > * Referring to Figure 2(d) when explaining the witness complex to improve this explanation. Rewording simplicial complexes "approximating the topology" to "approximating the homology" and removing the reference to a 'symplectic vertex'
> > * Removing "assumes topology τ" and clarifying that the model's learned manifold approximates the topology of the true data manifold (and superlevel sets of density of the true data manifold) through the learning process.
> > * Figure 3 has been updated to highlight the conditioning, and Figure 4 has been updated to clarify that it shows conditional submanifolds which were clustered together in one row and those which were not clustered together in the next row, and Figure 6 has been updated to correspond to our evaluation metric.
> > * The phrase “that best separates similar and dissimilar topologies” has been made more precise, based on your suggestion.
> > * Rewording "metric" to "dissimilarity measure" or "evaluation metric", avoiding ambiguity with the meaning of "metric" in the mathematical sense of distance or in the broader use of metric in evaluation.
> > * Replacing Greek letter ξ for Roman letter c when referring to number of biclusters, for consistency of Greek and Roman lettering
> > * Clarifying that submanifolds may have different homology (not "topology") than their supermanifolds
> > * Referring to "persistent homology" instead of "homology" where more appropriate. We mean persistent homology throughout the paper.
> > * Clarifying "reals" to mean real data, not R^n
> > * Incorporated each of your suggestions regarding style and clarity, including rewording "groups" as "clusters" where appropriate, as we are measuring topological similarity, not making claims on homeomorphic groups, thereby avoiding possible ambiguity with "homology groups" here.
> >
> > Thanks, again, for your detailed review in making our work much clearer and from there, impactful in moving work at this intersection forward.

---

> > > ### Comment · AnonReviewer5 · 2020-11-16
> > > **Thanks for the updates**
> > >
> > > Thanks for the detailed response and the care with which you addressed my concerns, I really appreciate it! I will read the updated version in detail and respond with any additional questions I might have.

---

> > > > ### Author Response · Authors · 2020-11-22
> > > > **Thanks and happy to answer more questions**
> > > >
> > > > Thank you so much. We are more than happy to answer all additional questions. Please let us know if it would be easier for you to see a version with changes highlighted; we can upload that instead for now. In the meantime, hope you find the revised portions satisfactory. Thanks again for the fortifying comments.

---

> ### Author Response · Authors · 2020-11-25
> **Author response to update: conciseness and algorithmic details**
>
> Thank you sincerely for your thoroughness in reviewing our revision and for supporting the theoretical basis of our work — we believe your suggestions have strengthened this paper and agree that the application of W. RLTs to measuring disentanglement is a key contribution. To that end we have modified the main text to better convey to the reader how to understand and apply our metrics. After making the paper a bit more concise by removing some repeated details, we have also moved a key algorithm for computing our metric to the main body (due to space constraints we were unable to shift all the algorithms). In the Experiments section we have added procedural information on taking a dataset and generative model of interest and computing our metrics using the appropriate algorithms, as well as relevant experimental parameters. We believe that these changes give the reader a more complete view of our application of TDA to disentanglement.

---

### Decision · Program_Chairs · 2021-01-07
**Final Decision**

**Decision:**

Accept (Poster)

**Comment:**

The paper proposes the use of topological similarity between conditional submanifolds for a given latent dimension as a metric for measuring disentanglement in generative models. To estimate the topological similarity between conditional submanifolds, the authors build upon an earlier work of Relative Living Times (RLT).

R5 and R4 had concerns on the paper, particularly about the lack of enough novelty in the actual technique (R5) and about the lack of convincing experiments (R4). One of the concerns raised by R4 was around the discrepancies between MIG and FactorVAE. However as noted by other reviewers (R2 and R5), these discrepancies between different popular metrics are well acknowledged in the literature and authors have responded to this point. R2 and R5 also appreciate that avoiding the rotation issue faced by most of these disentanglement metrics is one of the strengths of the proposed metric.

While I tend to agree with R5 that the actual technique is inspired from the earlier work on "Geometry Score", I also think the application to measuring disentanglement in generative models is a novel contribution in itself, especially because current metrics have issues as pointed out by other reviewers -- the paper provides a fresh conditional sub-manifold perspective on disentanglement and a theoretically sound metric for measuring disentanglement.

Considering this novel perspective and a resulting theoretically sound metric for measuring disentanglement that addresses some of the issues with current metrics, I recommend accepting the paper.